# Decoding Attention from Gaze:
# A Benchmark Dataset and End-to-End Models

**Karan Uppal**                                              KARAN.UPPAL3@GMAIL.COM
*Indian Institute of Technology, Kharagpur, India*

**Jaeah Kim**                                                JAEAHK@ANDREW.CMU.EDU
*Carnegie Mellon University, Pittsburgh, USA*

**Shashank Singh**                                           SHASHANKSSINGH44@GMAIL.COM
*Max Planck Institute for Intelligent Systems, Tübingen, Germany*

**Editor:** Editor's name

## Abstract

Eye-tracking has potential to provide rich behavioral data about human cognition in eco-
logically valid environments. However, analyzing this rich data is often challenging. Most
automated analyses are specific to simplistic artificial visual stimuli with well-separated,
static regions of interest, while most analyses in the context of complex visual stimuli, such
as most natural scenes, rely on laborious and time-consuming manual annotation. This
paper studies using computer vision tools for "attention decoding", the task of assessing
the locus of a participant's overt visual attention over time. We provide a publicly available
Multiple Object Eye-Tracking (MOET) dataset, consisting of gaze data from participants
tracking specific objects, annotated with labels and bounding boxes, in crowded real-world
videos, for training and evaluating attention decoding algorithms. We also propose two end-
to-end deep learning models for attention decoding and compare these to state-of-the-art
heuristic methods.

**Keywords:** Gaze; Eye-Tracking; Deep Learning; Attentional Decoding

## 1. Introduction

Eye-tracking data have been shown to provide rich information about diverse cognitive
functions, including spatial cognition (Kiefer et al., 2017), attention (Kim et al., 2021, 2022),
reading and language acquisition (Eng et al., 2020; Kim and Grüter, 2021), counterfactual
simulation (Gerstenberg et al., 2017), and emotion (De Lemos et al., 2008; Tep et al.,
2008). These data are thus collected in many areas of cognitive science and human behavior
research (Tanenhaus and Spivey-Knowlton, 1996), as well as in marketing (Chandon et al.,
2006; Zamani et al., 2016; Wedel and Pieters, 2008, 2017) and human-computer (Jacob and
Karn, 2003; Granka et al., 2004; Majaranta and Bulling, 2014) and human-robot (Admoni
and Scassellati, 2017; Aronson et al., 2021; Aronson and Admoni, 2022) interaction research.
However, analyzing these rich data can be challenging. Specifically, in most experimental
paradigms using eye-tracking, a key step of data analysis involves determining which, out of
several possible loci in the participant's visual environment, the participant is attending to,
at each point in time. In this paper, we call this step "attention decoding", since it involves
decoding a participant's latent attentional state from their measured gaze behavior.

Most research using eye-tracking takes one of two approaches to attention decoding,
depending on the complexity of the participant's visual environment. On one hand, in

simple visual environments that include only a small number of well-separated objects on a static display, simple automatic measures, such as the number and durations of fixations within predetermined regions of interest (ROIs), can suffice to determine the locus of the participant's attention. As examples, this approach has been used by papers studying reading comprehension (Carpenter and Just, 1977; Tsai et al., 2012; Raney et al., 2014; Eng et al., 2020) or using the "visual world paradigm" (Mirman et al., 2008; Huettig et al., 2011; Kim and Grüter, 2021). Beyond these simple visual settings, the vast majority of eye-tracking data analysis involves training human coders to classify gaze-behavior frame-by-frame or fixation-by-fixation (Steinbach, 1969; Tsang et al., 2010; Kurzhals et al., 2014b; Vansteenkiste et al., 2015; Fraser et al., 2017; Miller et al., 2020; Kim et al., 2020; Pellicer-Sánchez, 2016). This second approach has become especially prevalent with the recent advent of cheaper and more reliable head-mounted eye-trackers and video cameras, which enable flexible data collection from a first-person perspective (Franchak et al., 2011; Kurzhals et al., 2016; Bambach et al., 2018).

Both of these approaches to attention decoding have contributed to a rich body of eye-tracking-based research, but both approaches have notable limitations. The first approach is limited to very simple settings that fail to emulate the visual complexity of the real world. Meanwhile, despite extensive work on developing software for efficient manual coding (Tsang et al., 2010; Kurzhals et al., 2014b), training human coders to hand-code attentional state frame-by-frame is typically labor-intensive and slow, and may be error-prone or subjective, impacting reproducibility of scientific results (Kim et al., 2020). This paper explores a different approach: using recent tools from machine learning to automate attention decoding in rich, dynamic visual contexts. A robust automatic attention decoding algorithm would enable a fast, accurate, and reproducible approach to analyzing these data, significantly accelerating eye-tracking-based behavior research.

One challenge in developing such an algorithm is the lack of labeled data for training and evaluating learning-based algorithms. Hence, the present paper has two main contributions. First, we present a publicly available dataset, called the Multiple Object Eye-Tracking (MOET)[1] dataset, which consists of eye-tracking data collected from human participants viewing several dynamic natural scene videos, as well as class labels and bounding boxes for objects participants were assigned to track on each frame of the videos. This dataset can serve as a benchmark for researchers working on improving attention decoding methods. Second, we design and evaluate two end-to-end attention decoding algorithms and compare their performance to that of a state-of-the-art attention decoding algorithm based on applying heuristics to the output of an off-the-shelf object detector.

## 2. Related work

### 2.1. Attention decoding methods

State-of-the-art methods for automated attention decoding typically take a two-step, frame-by-frame approach that separates the tasks of object detection and gaze processing and

---

1. This name is a reference to the Multiple Object Tracking task used *in machine learning* (Milan et al., 2016) and should not be confused with the Multiple Object Tracking task *in psychology and vision research*, a behavioral paradigm widely used to study visual attention (Pylyshyn and Storm, 1988; Meyerhoff et al., 2017).

produces a decoding prediction for each frame using only stimulus and gaze data from that frame. Most commonly, the stimulus video is first passed through an off-the-shelf object detector to obtain bounding boxes for all objects in each frame, and then these bounding boxes and the participant's gaze location are used to determine the locus of attention according to deterministic rules. For example, Kumari et al. (2021) used YOLOv4 (Alexey et al., 2020) to detect objects and then considered the participant to be attending to an object if the gaze coordinates fell within that object's bounding box. Machado et al. (2019) used MobileNet (Howard et al., 2017) to identify the objects in each frame, and then considered the participant to be attending to the object whose bounding box contained the greatest number of fixations within a short time window (with certain additional heuristics to handle overlapping objects). Panetta et al. (2019) and Rong et al. (2022) take a slightly different two-step approach, first extracting a small region of the image near the participant's gaze and then passing that region through an object detector or classifier. To improve results for irregularly shaped objects, Wolf et al. (2018) and Deane et al. (2022) used segmentation masks from mask R-CNN (instead of bounding boxes) and then considered the participant to be attending to an object if their gaze lay within the corresponding mask.

These methods can perform well when the possible loci of attention in the scene are well-separated, but they struggle when objects are crowded or can occlude each other. In such settings, accurate attention decoding requires aggregating gaze information across multiple frames; for example, Kim et al. (2020) do this (in an artificial visual setting with simple moving shapes) using Markov models of attentional state over time. In the present work, we attempt to overcome this challenge with end-to-end attention decoding architectures that use recurrent layers to combine gaze data from multiple consecutive frames.

End-to-end attention decoding requires a way to incorporate participants' gaze data into the decoding network while retaining its spatiotemporal structure. To do this, we utilize a *gaze pooling layer* (originally proposed by Sattar et al. (2020) for the distinct but related task of visual search target prediction), which acts as an attention mechanism within the neural network, selecting visual features based on their proximity to the participant's gaze. Another viable approach might be that of Sims and Conati (2020), who present a deep learning model that directly takes in the entire eye-tracking dataset as input (using both convolutional and recurrent layers, in two separate feature paths, to capture spatial and temporal features of gaze). However, their task (user confusion detection) involves making a single prediction for an entire video, and it is not clear how to adapt their method to make predictions for every frame of the video.

## 2.2. Datasets for attention decoding

Data used to train and evaluate most attention decoding methods, including those used by Machado et al. (2019); Panetta et al. (2019); Kumari et al. (2021) and Deane et al. (2022), do not appear to be publicly available. The training data of Wolf et al. (2018) is available, but is very small (72 images, with 2 object classes). Rong et al. (2022) (who only classified the attentional locus and did not predict a bounding box) used the publicly available Dr(eye)ve dataset (Palazzi et al., 2018), which does not contain bounding box annotations. Consequently, there is a lack of large labeled datasets for training and evaluating attention decoding methods. One of the main contributions of this paper is therefore to provide a

publicly available dataset, called the Multiple Object Eye-Tracking (MOET) dataset, which consists of eye-tracking data collected while human participants viewed a subset of videos from the Multiple Object Tracking 2016 (MOT16) benchmark dataset (Milan et al., 2016), as well as labels and bounding boxes for objects participants were assigned to track on each frame of the videos. We hope this dataset will serve as a benchmark for researchers working on improving attention decoding methods.

The most similar available dataset appears to be the VISUS dataset (Kurzhals et al., 2014a), whose structure is comparable to our dataset, including gaze data from 25 participants watching 11 different videos. There are a few notable differences between the two datasets. First, the videos in the VISUS dataset are less crowded; some of the videos have only a single salient object; in contrast, frames in the MOT16 dataset contain an average of 19.15 detectable objects (Milan et al., 2016). Second, 9 of the 11 VISUS videos are recorded from a static camera; in contrast, 8 of the 14 videos in the MOT16 dataset are taken from moving perspectives. The latter better reflect, for example, the visual features of first-person (head-mounted) eye-tracking data. Finally, in the VISUS dataset, rather than tracking a specific cued Target, participants were assigned task to perform while watching each video (e.g., count the number of times a ball is passed). The tasks were designed to induce particular patterns of gaze behavior. Although this design may lead to more naturalistic viewing behaviors than assigning randomly selected target objects, it has several drawbacks. Assigning a task, rather than a specific target, results in some ambiguity regarding the true locus of attention. In some frames, the same task might correspond to multiple different viewing targets, while many frames lack a clear Target object (e.g., when the participant is searching for an object that has not yet appeared). Partly for this reason, the VISUS dataset does not include annotated attentional loci, and so training an attention decoding model requires one to manually annotate the expected Target in each frame where a target locus can be inferred from the task description. Also, since the tasks are the same for all participants (except for 3 of the VISUS videos, for which each participant was assigned one of two different tasks), there is far less diversity in viewing behaviors and attentional loci across participants the dataset. For these reasons, we believe our MOET dataset may be better suited to training and evaluating models for attention decoding.

## 3. The Multiple Object Eye-Tracking (MOET) dataset

**Overview**   Here, we present the Multiple Object Eye-Tracking (MOET) dataset, publicly accessible on the Open Science Framework at https://osf.io/28rnx/. MOET is an extension of the Multiple Object Tracking 2016 (MOT16) benchmark dataset (Milan et al., 2016, https://motchallenge.net/data/MOT16/), a dataset of 14 videos (11 221 frames totaling 7 min, 43 s in length, mean 33.0 s per video, std. dev. 18.9 s) widely used for training and evaluating multiple object tracking algorithms. MOET extends MOT16 with eye-tracking data obtained from 16 participants viewing the 14 videos while tracking distinct target objects through the videos. Target objects are annotated in each frame with class labels and bounding boxes, providing a total of $16 \times 11221 = 179536$ annotated frames. After removing frames with missing gaze data or where the participant's gaze deviated significantly from the target (as described in Section 3.2), the dataset contains a total of 105532 labeled frames. Moreover, to ensure the data contain a variety of attentional loci and plenty of

attentional transitions between different loci, the assigned target object frequently changed (on average, every 2.675 s, roughly 10 times per video; see "Target Selection Procedure" below for details). Some summary statistics describing our MOET dataset are provided in Table 9 of Appendix D.

**MOT16 videos** The MOT16 dataset Milan et al. (2016) is a widely used benchmark dataset for multiple object tracking. We chose to build the MOET dataset on top of the MOT16 dataset because the latter contains a large number of crowded moving objects in complex real-world scenes, making it a realistic and challenging environment for attention decoding. This contrasts with the scenes used in previous work on attention decoding, which have typically been laboratory environments with a relatively small number of salient objects towards which attention might be directed. Table 10 of Appendix D provides summary statistics regarding the MOT16 videos, as well as a summary of the most common objects detected in the MOT16 datasets. Further details regarding the MOT16 dataset can be found in Milan et al. (2016).

### 3.1. Data collection

**Target Selection Procedure:** For each stimulus video and participant, we randomly generated a sequence of targets, which were indicated by a green ellipse (inscribed into the object's bounding box) as the participant viewed the video. The sequence of target objects was generated according to Algorithm 1, which was designed to ensure that participants transitioned frequently between objects. First, we used an off-the-shelf object detector (RetinaNet (Lin et al., 2017b), trained on the Microsoft "Common Objects in COntext" (COCO) database (Lin et al., 2014)) [2] The first target object was selected randomly from those present in the first frame of the video; to prevent the target from switching too often, we selected objects with probabilities proportional to the number of frames they would remain in the video. That object remained the target either until it disappeared (due to occlusion or leaving the frame) or until a random "switch time" (whichever occurred first). "Switch times" were exponentially distributed with minimum 1 s and mean 2.5 s. We then repeated this procedure to choose the next target object, until target objects were selected for the entire video.

Participants viewed each of the 14 MOT16 videos 3 times, using different confidence thresholds (40%, 60%, and 80%) for the target assignment object detector. This was done in order to ensure a balance between the confidence of the object detections and the density of candidate target objects in the videos. In this paper, we only analyze data from the 60% confidence condition, but the public dataset includes data at all three thresholds.

**Task** Participants were asked to watch a series of videos and to track the target object, indicated by a green ellipse, as continuously as possible, with their eyes. The order of the 42 videos was randomized for each participant. After each video, participants were given a 5 second break during which a black screen was displayed. The experimental protocol was approved by the Carnegie Mellon University Institutional Review Board, and informed consent was obtained from each participant before the study.

---

2. See http://cocodataset.org/#explore for a list of COCO object classes and exemplars from each class.

---

**Algorithm 1:** Procedure for generating viewing targets.

---

**Input:** Stimulus video `stim_video`
**Output:** List of assigned targets throughout the video
`detected_objects ← object_detector(raw_video)`
`/* detected_objects is of type list[list[(object, duration)]] and`
` contains, for each video frame, a list of detected objects together`
` with their remaining duration in the video */`
$T ← $ `length(stim_video)`
$t ← 0\,\text{s}$
`target_list ← []`
**while** $t < T$ **do**
    `// sample target with probability proportional to remaining duration`
    `target ← sample(detected_objects[t])`
    $\Delta t ← \min\{\texttt{target.duration}[t],\ 1\,\text{s} + \text{Exp}(0.66\,\text{s})\}$
    $t ← t + \Delta t$ `// next transition time`
    `target_list.append((Target, `$t$`))`
**end**
`return target_list`

---

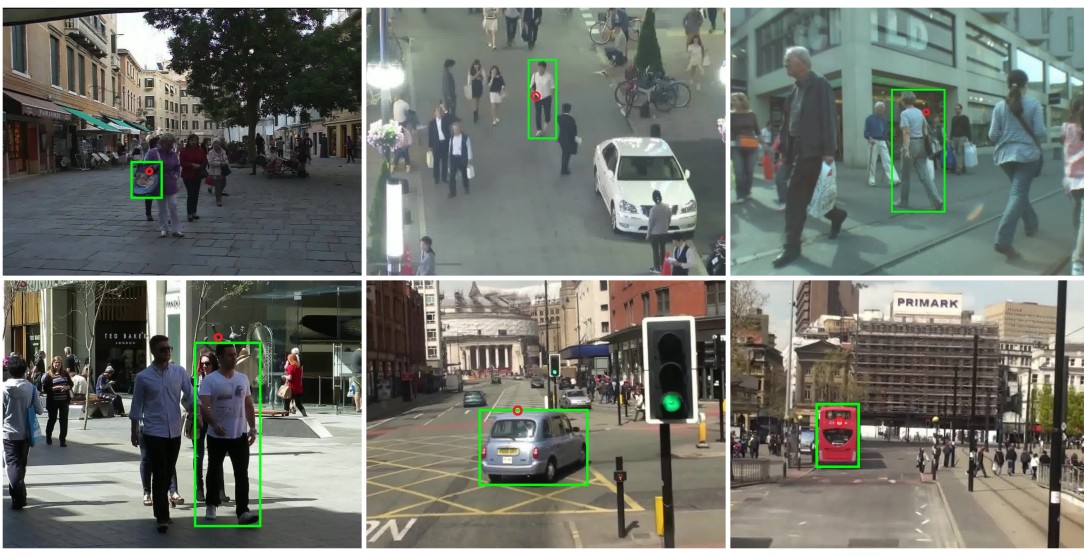

Figure 1: Example frames from stimulus videos in the experiment. The the target object is indicated by a green bounding box and the gaze point is indicated by a red circle.

**Eye-tracking and display setup** While participants performed the task, their gaze was recorded at 60 Hz using an SMI RED-250 mobile eye-tracker SMI (2009). Prior to the experiment, the eye-tracker was calibrated to the participant using the 5-point calibration procedure in SMI's iView X software (with default settings). This involved participants

fixating for at least 400 ms on each of 5 fixation points distributed over the display. After calibration, we assume gaze was spatially aligned to the stimulus. Since gaze was recorded by the same computer presenting the stimulus, the stimulus and gaze data were synchronized automatically. Participants were seated at a distance of $\approx 50$ cm from the desktop display, which had physical dimensions 52 cm $\times$ 33 cm ($\approx 54.95° \times 36.53°$ visual angle) and pixel dimensions 1920 px $\times$ 1200 px, from which participants were seated at a distance of $\approx 50$ cm.

### 3.2. Data cleaning & preprocessing

Our experimental paradigm relies on the assumption that participants are tracking the assigned target object, so that the assigned target objects can be used as labels for attention decoding. Because, in reality, participants track the target imperfectly, these ground truth labels contain a significant amount of noise. Hence, cleaning the data is paramount before proceeding to training. We performed the following data-cleaning steps:

1. **Dropping transition frames:** The target object the participant was assigned to track changed frequently over the experiment. Since humans typically take 200-300 ms to saccade to an exogenous visual cue (Purves et al., 2001; Palmer et al., 2019), we omitted frames for 300 ms after each target transition, leaving 162205 frames.

2. **Imputation of missing data:** When gaze data was missing for short ($< 10$ frames $\approx 167$ ms) sequences of frames (e.g., due to the participant blinking), we interpolated the gaze position during these frames linearly from the gaze positions in adjacent non-missing frames. Remaining frames with missing gaze data were omitted from the dataset. We also omitted a small number of frames in which no objects were detected, leaving 124658 frames.

3. **Verifying overt condition:** Our initial assumption is that the participants gaze is overt, that is, their gaze is directed at the object to which they are attending. However, we found that participants did not always direct their gaze towards the specified target object (perhaps due to distraction or fatigue, or because they transitioned slowly to a new target). We therefore applied a distance threshold, dropping frames in which gaze was more than a certain Euclidean distance (100 px $\approx 3.1°$ visual angle) away from (the nearest point of the bounding box of) the assigned target object. This left 105532 frames for final analysis.

## 4. Methods

### 4.1. End-to-end attention decoding architectures

In this section, we propose two end-to-end attention decoding models, illustrated in Figure 2. Both models use a pretrained object detector backbone and incorporate gaze using gaze density maps. The two models then differ in how they incorporate data across multiple consecutive frames; one uses a gated recurrent unit (GRU), which the other simply uses a fully connected layer.

**Backbone:** The model consists of a ResNet-50 Feature Pyramid Network (FPN) (Lin et al., 2017a) backbone, extracted from a Faster R-CNN (Ren et al., 2015) model pretrained on the MS COCO dataset (Lin et al., 2014). This is used to generate a feature map $F(I)$ for each input frame $I$. As we are interested in combining gaze features with the image features, we used $F(I)$ such that it still has a spatial resolution; specifically, we used the

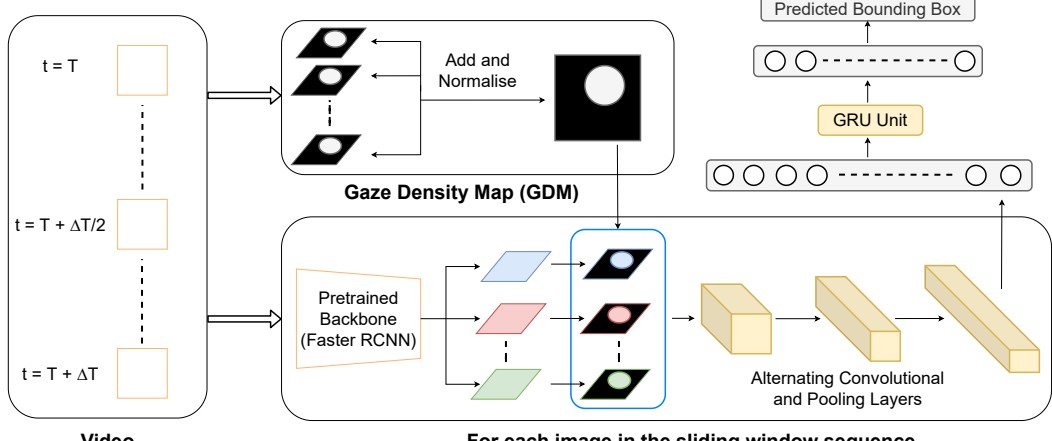

Figure 2: GRU Model Architecture

highest-resolution feature map ($128 \times 128$) output by the FPN. The weights of this backbone network are frozen during training of our models, as described below.

**Gaze Density Map (GDM):** Suppose that, within a time interval $[T, T + \Delta T]$, a participant $P$ gazes at a sequence $G(P, T) = \{(x_i, y_i)\}_{t \in [T, T+\Delta]}$ of screen coordinates. Taking inspiration from (Sattar et al., 2020), we aggregate this gaze sequence into a gaze density map, GDM($G$), in order to capture their spatial density over time. The gaze density map GDM($g$) for a single gaze point $g \, \epsilon \, G(P, T)$ is represented by a Gaussian, centered at the coordinates of the gaze:

$$\mathrm{GDM}(g) = \mathcal{N}(g, \sigma), \tag{1}$$

where the standard deviation $\sigma$ is a model hyperparameter. The gaze density map for all fixations GDM($G$) over a time interval $[T, T + \Delta T]$ is obtained by the coordinate-wise summation, normalized by the max value:

$$\mathrm{GDM}(G) = \frac{\sum_{g \, \epsilon \, G(P,T)} \mathrm{GDM}(g)}{\max \sum_{g \, \epsilon \, G(P,T)} \mathrm{GDM}(g)} \tag{2}$$

This can be interpreted as a temporally localized feature importance map, describing where the participant is visually attending to during the time interval $[T, T + \Delta T]$.

**Incorporating Gaze:** We combine the image features with the gaze data in a weighting mechanism. GDM($G$) is downsampled to match the resolution of $F(I)$. This integration is performed by an element-wise multiplication between both to obtain a weighted feature map $F_{wgt}(I)$:

$$F_{wgt}(I) = F(I) \otimes \mathrm{GDM}(G) \tag{3}$$

The weighted feature map is then passed through 2 alternating 3x3 convolutional and 2x2 pooling layers, doubling the depth at each step. Finally, we add a fully connected layer on top of this to create a 512-dimensional vector.

**Aggregating Temporal Information:** We expected dealing with cases of object collision/occlusions requires aggregating information across multiple frames of the data. In order to combat such cases, we add a GRU unit with a single hidden layer which takes

the 512-dimensional vector as input for each input frame in the time interval $[T, T + \Delta T]$. The output is further passed to a fully connected layer which predicts the bounding box coordinates for each input frame in time interval $[T, T + \Delta T]$. The model architecture is illustrated in Fig. 2. For an input frame $T$, we thus have up to $\Delta T$ predictions from the batches corresponding to time intervals $[T - \Delta T, T], [T - \Delta T + 1, T + 1], ..., [T, T + \Delta T]$. These predictions are averaged coordinate-wise to generate the final predicted bounding box. We refer to this model as the **GRU Model**.

The above model utilizes both spatial and temporal information. We also evaluate another model having only the spatial component. Taking $\Delta T = 1$, we replace the GRU unit with a fully connected layer, thereby considering only one input frame at a time. We refer to this model as the **Feedforward Model**.

### 4.2. Training procedure

The input frames are obtained in a sliding window approach. Considering a time interval $[T, T + \Delta T]$ and $N$ to be the total number of frames, we extract video frames from the time interval: $[T, T + \Delta T]$, for each $T \in \{0, 1, ..., N - \Delta T\}$, in the form of batches.

Each input batch is passed to the model described above, which predicts $T$ bounding box coordinates, corresponding to each frame. The dataset is split across participants, with 20% belonging to the validation set and 80% belonging to the training set. Each model is trained for 30 epochs, using $L_1$ loss and Adam for stochastic optimization. Training hyperparameters are listed in Table 1. Appendices A and B respectively provide further details on the selection of the $\sigma$ hyperparameter and the computational infrastructure used to train models.

| Hyperparameter | Values Considered |
|---|---|
| Learning rate | 0.001 |
| Weight Decay | 0.0001 |
| GDM Standard deviation ($\sigma$) | 100, 250, **500**, 750, 1000 |
| Time interval ($\Delta T$) | **1, 5** |

Table 1: Hyperparameters of end-to-end models. Final values used in evaluation are in bold.

### 4.3. Evaluation procedure

We compare our end-to-end neural network approach to several heuristic baselines:

1. **Fixed Box Baseline:** For each video, the fixed-box baseline predicts a bounding box of fixed size centered around the gaze point. The size of the bounding box is computed as the average size of all bounding box labels, across all participants, for that video. This approach is similar to those of Panetta et al. (2019) and Rong et al. (2022), which select regions of interest based only on gaze.

2. **Object Detector (OD) Baseline:** Using the pretrained Faster RCNN model, candidate bounding box predictions are generated for each input frame. If the gaze point is

present inside any of the candidate bounding boxes, it is chosen to be the object of interest. In the case of overlapping bounding boxes, the one with the highest prediction probability (as given by the Faster RCNN model) is chosen. If the gaze point does not lie inside any of the predicted bounding boxes, then a random candidate is chosen to be the object of interest.

3. **Object Detector (OD) Mod:** It is similar to the above baseline with a slight modification in the case when the gaze point doesn't lie inside any of the candidate bounding boxes. In such a scenario, the nearest bounding box is chosen to be the object of interest, where "nearest" is defined in terms of the Euclidean distance between the gaze point and the nearest pixel of the bounding box. This is essentially the method proposed by (Kumari et al., 2021), and also used (with segmentation masks rather than bounding boxes) by Wolf et al. (2018) and Deane et al. (2022).

4. **Object Detector (OD) Oracle:** The "OD Oracle" model is given access to the true target bounding box and selects the best bounding box out of those provided by the object detector; while this is not feasible in practice, the performance of this model provides a reference upper bound on the best performance possible by any model selecting one of the bounding boxes output by the object detector.

The models are evaluated on the basis of mean (across frames) Intersection over Union (IoU) between the predicted and ground truth (assigned target) bounding boxes.

## 5. Experiments results

We present two sets of results: the first involves studying generalization across data from different participants (watching the same videos), while the second involves studying generalization across data collected while the same participants watched different videos. Code for reproducing our results is available on GitHub at https://github.com/karan-uppal3/decoding-attention.

### 5.1. Generalization across participants

In some applications, we may have labeled data from some participants watching a video and want to apply the model to data from new participants watching the same video. This is useful, for example, for accelerating annotation of datasets in which many participants watch the same videos.

To evaluate generalization across participants in this manner, for each video, we trained each model on data 75% of the participants (12 participants) and then evaluated performance on the remaining 25% of participants (4 participants). Average results across all 14 videos, given in Table 2, suggest that the best results were achieved by the OD Baseline and OD Mod methods, which apply heuristics to combine gaze with the output of an off-the-shelf object detector. However, the much higher performance of the OD Oracle also suggests that there may be substantial room for improving performance over these heuristic methods.

In this setting, one open question is whether it is better to train only on the target video, or whether training on a larger dataset including other videos improves performance. However, training a model to convergence on a large number of videos is also computationally

| Approach | Algorithm | Mean IoU (Std. Dev.) |
|---|---|---|
| Completely Rule-Based | Fixed Box Baseline | 0.244 (0.060) |
| Object Detector (OD) + Rule | OD Baseline | 0.495 (0.112) |
| | OD Mod | 0.546 (0.089) |
| | OD Oracle | 0.717 (0.077) |
| End-to-End (Ours) | Feedforward Model | 0.443 (0.116) |
| | GRU Model | 0.406 (0.153) |

Table 2: Means and standard deviations (across 16 participants) of mean IoUs (across 14 videos), for each algorithm. Faded values indicate cases where the model uses "oracle" knowledge of the true label and are provided only for comparison.

rather challenging. Hence, to test this, we also trained the Feedforward model collectively on training data from only the *first four* MOET videos (shuffled together), using the same number of epochs (corresponding to 4× as many total gradient updates). We then evaluated this model on the test subset of each of the four videos. As shown in Table 3, for Videos 1 and 2, the models trained on individual videos significantly outperformed the model trained on all 4 videos, while the difference in performance between the two models was not significant for Videos 3 and 4. This suggests that it may be best to train a separate model for each video being used at test time, although further work with higher-capacity neural networks might be able to construct an effective unified model for this problem.

| Feedforward Model | Video 1 | Video 2 | Video 3 | Video 4 |
|---|---|---|---|---|
| Videos 1-4 | 0.302 (0.034) | 0.335 (0.085) | **0.450** (0.071) | 0.395 (0.063) |
| Individual Videos | **0.513** (0.054) | **0.469** (0.053) | 0.416 (0.095) | **0.432** (0.096) |

Table 3: Means and standard deviations (across 16 participants) of mean IoUs, comparing the performance of joint training with individual models.

## 5.2. Generalization across videos

In other applications, we may have data from participants watching some videos and want to apply the model to data collected while watching *different* videos. We expect this to be more challenging than generalizing across different participants watching the same video, because, whereas the relationship between gaze and attention is likely to be similar across individuals, the distribution of features across videos can vary widely.

More specifically, we expected transfer of learning across videos to depend strongly on the similarity between features of the training and test videos. Hence, to study this, we divided videos into training and testing videos in two ways. First, the MOT16 videos are given in pairs of videos with similar high-level features. For example, Videos 3 and 4 both consist of elevated views of a crowded shopping street at night, while Videos 13 and 14

were both filmed from aboard moving buses at busy intersections. Hence, we considered models trained on one element of each pair and tested on the other. Second, one might expect models trained on several distinct videos to generalize better to new videos. Hence, we trained a model on MOET Videos 1-4 (as in the previous section) and evaluated them on other remaining videos. Table 4 shows the performance of these models on each test video.

| Training video(s) | Test video | | | | | | | |
|---|---|---|---|---|---|---|---|---|
| | Video 1 | Video 3 | Video 5 | Video 7 | Video 9 | Video 11 | Video 13 | Mean |
| Pairs | 0.170 | 0.205 | 0.329 | 0.076 | 0.079 | 0.419 | 0.082 | 0.1942 |
| 1-4 | 0.302 | 0.450 | 0.062 | 0.163 | 0.200 | 0.109 | 0.088 | 0.1243 |

Table 4: Mean (over 16 participants) validation IoU of the feed-forward end-to-end model trained and tested on different videos. Faded values indicate cases where the training and test videos overlap. For "Pairs", we trained the model on each $i \in \{2, 4, ..., 14\}$ and tested it on Video $i - 1$. For "1-4", the model was trained on Videos 1, 2, 3, and 4.

## 6. Limitations & future work

This section discusses some limitations of the MOET dataset that should be considered by users, as well as limitations of the machine learning analyses and methods reported in the main paper that might suggest avenues for future work.

**Limitations of the MOET dataset**   Although the MOET dataset is fairly rich compared to comparable existing datasets, a few factors limit its generalizability:

1. The target objects are limited to the 80 common categories in the COCO dataset (Lin et al., 2014). Moreover, in the MOT16 videos, by far the most common objects are of the class "Person", which was consequently the target object class on approximately 85% of frames. The MOT16 source videos are from fairly similar settings, primarily involving outdoor pedestrian or motor areas in large European and Asian cities. Developing robust attention decoding methods will require both a greater variety of detected objects as well as more diverse natural scenes.

2. The quality of frame labels depends on how accurately participants track the target objects. Although simple preprocessing steps such as those described in Section 3.2 can remove the majority of noisy frame labels, it is unclear whether these steps introduce unrealistic biases into the data.

3. The sequence of target objects was artificially generated, and hence higher-order viewing patterns (e.g., transition times and distances) may not reflect those of human free-viewing behavior.

4. This experiment involved participants viewing videos on a display. While this continues to be the most common setting in which eye-tracking data is collected, it is not clear whether models trained in this setting will generalize to data from head-mounted eye-trackers, which are also becoming increasingly common.

Despite the substantial manual labor involved in human annotators labeling such a dataset, it would be of substantial interest to also collect a "free-viewing" dataset in which participants freely view the videos (i.e., without a specified target object).

**Limitations of current methods & evaluation** In this study, we only analyzed models for the task of predicting the bounding box of the participant's attentional locus. Other closely related tasks, including labeling the attentional locus (Panetta et al., 2019; Rong et al., 2022) and detecting transitions of attention between loci (Kim et al., 2020) should also be studied; the MOET dataset should prove useful for studying these problems.

## 7. Conclusions

Automating attention decoding in a way that is robust to different visual environments and participant behaviors is a challenging problem, and the technology will continue to improve as the underlying computer vision technologies improve, as more labeled data for attention decoding becomes available, and as behavioral researchers experiment with and provide feedback on attention decoding tools. This paper is intended to facilitate next steps in this direction, by providing a benchmark dataset for attention decoding and by comparing two end-to-end attention decoding models with existing heuristic algorithms.

The end-to-end models we considered were generally outperformed by two-stage methods combining off-the-shelf object detectors with heuristic rules to select a bounding box from those output by the object detector. At the same time, heuristic methods performed far worse than the optimal "oracle" predictor, suggesting that there may be substantial room for improvement in attention decoding methods.

We also provided preliminary evidence for a few interesting hypotheses about the difficulty of generalization in attention decoding. First, it appears that generalization across videos is much more challenging than generalization across participants. We believe this may be because the differences in visual features between different videos are generally much greater than the differences in gaze patterns across different participants performing a similar task. Second, we found that a model trained and tested on a single video outperformed models that were trained on several videos including the test video. Both of these findings are consistent with the idea that our end-to-end models tend to overfit to specific visual features in each video, and new methods may be needed to encourage learning visual features that generalize across videos.

## Acknowledgments

We thank the Cognitive Development Lab (PI: Dr. Anna Fisher) and the Infant Language & Learning Lab (PI: Dr. Erik Thiessen) for support with study design and data collection. In particular, we thank Alice Russell and Varsha Shankar for helping to collect data. K.U. was supported by a DAAD WISE scholarship (Ref. no.: 91836256). J.K. was supported

by the US National Science Foundation (grant BCS-1451706). S.S. was supported by the German Federal Ministry of Education and Research (BMBF) through the Tübingen AI Center (FKZ: 01IS18039B).

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

## Appendix A. Hyperparameter search

As shown in Table 5, different values of $\sigma$ were optimal for different videos, and no single value of $\sigma$ clearly optimizes mean performance across videos.

## Appendix B. Computational Details

All the experiments were performed on Ubuntu 20.04.2 LTS, with implementation using PyTorch 1.11 and logging using the Weights & Biases (`wandb`) toolbox. The training time for each model depended on the length of the video(s). For individual videos, the GRU Model took an average training time of 20.68 hours while the Feedforward Model took an average training time of 14.57 hours, using a single NVIDIA Tesla V100 GPU.

| $\sigma$ | Video 1 | Video 3 | Video 5 | Video 7 | Video 9 | Video 11 | Video 13 | Mean (Std. Dev.) |
|---|---|---|---|---|---|---|---|---|
| 100 | 0.433 | 0.396 | 0.501 | 0.251 | 0.342 | 0.517 | 0.193 | 0.376 (0.122) |
| 250 | **0.546** | **0.499** | 0.488 | 0.266 | 0.298 | 0.539 | 0.200 | 0.405 (0.145) |
| 500 | 0.535 | 0.443 | 0.539 | 0.287 | 0.394 | 0.570 | 0.213 | 0.426 (0.136) |
| 750 | 0.496 | 0.359 | 0.560 | 0.268 | 0.390 | **0.574** | 0.269 | 0.417 (0.129) |
| 1000 | 0.536 | 0.406 | **0.574** | **0.324** | **0.397** | 0.545 | **0.272** | 0.436 (0.117) |

Table 5: Mean (over 16 participants) validation IoU of the feed-forward end-to-end model using different standard deviations $\sigma$ to compute the GDM. The largest mean IoU for each video is in bold.

## Appendix C. Additional Experimental Results

Table 6 provides the mean IoU for each video (averaged across all 16 participants) for the proposed baselines and end-to-end models.

| Videos | Fixed Box Baseline | OD Baseline | OD Mod | OD Oracle | Feedforward | GRU |
|---|---|---|---|---|---|---|
| 1 | 0.3280 | 0.4828 | 0.5354 | 0.7460 | 0.5130 | 0.4668 |
| 2 | 0.2790 | 0.5248 | 0.5530 | 0.7400 | 0.4690 | 0.4199 |
| 3 | 0.2854 | 0.4295 | 0.5003 | 0.6956 | 0.4159 | 0.2434 |
| 4 | 0.2740 | 0.4449 | 0.5556 | 0.6689 | 0.4316 | 0.2259 |
| 5 | 0.3248 | 0.5752 | 0.5902 | 0.6886 | 0.6254 | 0.6990 |
| 6 | 0.2388 | 0.3308 | 0.3710 | 0.5224 | 0.4065 | 0.4639 |
| 7 | 0.2343 | 0.4575 | 0.5081 | 0.6757 | 0.3465 | 0.2778 |
| 8 | 0.2607 | 0.6267 | 0.6515 | 0.8124 | 0.5685 | 0.4481 |
| 9 | 0.2508 | 0.5865 | 0.6046 | 0.7720 | 0.3643 | 0.4034 |
| 10 | 0.1493 | 0.4097 | 0.4593 | 0.6339 | 0.2991 | 0.3464 |
| 11 | 0.2659 | 0.6765 | 0.6875 | 0.7778 | 0.6407 | 0.6185 |
| 12 | 0.2249 | 0.6395 | 0.6522 | 0.7924 | 0.4944 | 0.5710 |
| 13 | 0.1654 | 0.3976 | 0.5397 | 0.7806 | 0.3557 | 0.2669 |
| 14 | 0.1334 | 0.3414 | 0.4291 | 0.7393 | 0.2662 | 0.2137 |
| Mean | 0.2439 | 0.4945 | 0.5455 | 0.7174 | 0.4426 | 0.4060 |
| Std. Dev. | 0.0595 | 0.1118 | 0.0892 | 0.0772 | 0.1156 | 0.1532 |

Table 6: Means and standard deviations (across 16 participants) of mean IoUs, of all proposed models. Faded values indicate cases where the model uses "oracle" knowledge of the true label and are provided only for comparison.

### C.1. Using Different Losses

Although we trained our end-to-end models using $L_1$ loss, one should consider using more sophisticated losses, such as Generalized IoU (GIoU; (Rezatofighi et al., 2019)) and Complete

IoU (CIoU; (Zheng et al., 2020)), that have been shown to significantly improve performance for many bounding box regression tasks. Table 7 provides the mean validation IoU for the feedforward model trained using GIoU loss and CIoU loss, across videos.

| Loss | Video 1 | Video 3 | Video 5 | Video 7 | Video 9 | Video 11 | Video 13 | Mean (Std. Dev) |
|------|---------|---------|---------|---------|---------|----------|----------|-----------------|
| $L_1$ | 0.5130 | 0.4159 | 0.6254 | 0.3465 | 0.3643 | 0.6407 | 0.3557 | 0.4659 (0.1274) |
| GIoU | 0.5667 | 0.5281 | 0.7210 | 0.2803 | 0.4479 | 0.6615 | 0.4279 | 0.5191 (0.1493) |
| CIoU | 0.5663 | 0.4312 | 0.7338 | 0.3993 | 0.4359 | 0.6912 | 0.4547 | 0.5303 (0.1355) |

Table 7: Mean (over 16 participants) validation IoU of the feed-forward end-to-end model using different loss functions.

### C.2. Average Gaze Density Map

As shown in Eq. 2, the Gaze Density Map is normalized to scale it to the interval $[0, 1]$. We experiment with a different variant of the GDM wherein we take the average of the provided maps:

$$\text{GDM}(G) = \frac{1}{\Delta T} \sum_{g \, \epsilon \, G(P,T)} \text{GDM}(g) \qquad (4)$$

The intuition is that the importance over the whole image is summed to one, rather than the most important pixel having a value of 1. Table 8 compares the mean validation IoU for the GRU model for the two variants of the Gaze Density Map. The performance of both is nearly identical.

| GDM | Video 1 | Video 3 | Video 5 | Video 7 | Video 9 | Video 11 | Video 13 | Mean (Std. Dev.) |
|-----|---------|---------|---------|---------|---------|----------|----------|------------------|
| Normalized | 0.4868 | 0.2434 | 0.6990 | 0.2778 | 0.4034 | 0.6185 | 0.2669 | 0.4280 (0.1809) |
| Averaged | 0.5220 | 0.2587 | 0.6994 | 0.3751 | 0.3641 | 0.4990 | 0.2461 | 0.4278 (0.1647) |

Table 8: Mean (over 16 participants) validation IoU of the GRU end-to-end model for different variants of Gaze Density Map.

## Appendix D. Dataset Summary Statistics

Table 9 provides summary statistics regarding the experimental setup and gaze data collected in our MOET dataset.

Table 10 provides summary statistics regarding the MOT16 videos used in our study, as well as the most common objects in the videos.

| Feature | Total | Mean (/video) | Mean (/participant) |
|---|---|---|---|
| Total frames | 179536 | 12824.0 | 11221 |
| Non-transition frames | 162205 | 11586.1 | 10137.8 |
| Non-transition frames w/ gaze | 124658 | 8904.1 | 7791.1 |
| Non-transition frames w/ gaze near target | 105532 | 7538.0 | 6595.8 |
| Distinct target objects | 2461 | 175.8 | 153.8 |
| Transitions between target objects | 2237 | 159.8 | 139.8 |

Table 9: Summary statistics of experimental conditions and gaze data in our MOET dataset.

| Feature | Number of videos | | | |
|---|---|---|---|---|
| Camera | Static: 6 | | Moving: 8 | |
| Viewpoint | Low: 1 | Medium: 9 | High: 4 | |
| Resolution | 1920 × 1080: 12 | | 640 × 480: 2 | |
| Lighting Conditions | Sunny: 4 Cloudy: 2 | Indoor: 2 | Night: 3 Shadow: 1 | |
| Framerate (FPS) | 30: 10 | 25: 2 | 14: 2 | |

| Most common objects | Detected instances | Distinct detected instances | Prop. frames present |
|---|---|---|---|
| Person | 107379 | 1137 | 99.8% |
| Car | 6377 | 72 | 34.8% |
| Handbag | 2110 | 120 | 15.9% |
| Motorcycle | 2023 | 19 | 17.6% |
| Bicycle | 1719 | 54 | 14.8% |
| Bus | 1112 | 10 | 5.9% |
| Traffic Light | 1066 | 29 | 6.5% |
| Chair | 984 | 21 | 4.6% |
| Potted Plant | 540 | 9 | 4.8% |
| Bench | 530 | 32 | 4.1% |

Table 10: Summary statistics of videos and most common objects detected in the MOT16 dataset.

## Appendix E.  Class-Wise Model Performance

To determine how much decoding performance varied based on the nature of the object being tracked, we evaluated the performance of the Feedforward model separately for each of the 10 most common objects. The results are shown in Table 11.

Although further work is needed to confirm such hypotheses, the results appear to suggest that performance is higher for larger objects (e.g., Car, Motorcycle, Bus) than for smaller objects (e.g., Bench, Handbag, Traffic Light), and perhaps that performance is higher for static objects (relative to the camera; e.g., a chair in Video 1 and a parked Car

UPPAL KIM SINGH

in Videos 3 and 4) than for quickly moving objects (e.g., Traffic Lights in Videos 13 and 14).

| Most common objects | Mean IoU |
|:---:|:---:|
| Person | 0.4123 |
| Bicycle | 0.3521 |
| Car | 0.5245 |
| Motorcycle | 0.5028 |
| Bus | 0.5334 |
| Traffic Light | 0.1610 |
| Bench | 0.1894 |
| Handbag | 0.2195 |
| Chair | 0.6456 |
| Potted Plant | 0.2051 |

Table 11: Class-wise performance of the Feedforward model on the 10 most common object classes, averaged over all 16 participants and 14 videos.

