# OpenReview forum: "Decoding Attention from Gaze: A Benchmark Dataset and End-to-End Models"
_NeurIPS.cc/2022/Workshop/GMML — Gaze Meets ML 2022 Poster_

### Official Review · Reviewer_EeaX · 2022-10-17
**Good quality benchmark dataset on gaze tracking.**

**Rating:** 5
**Confidence:** 4

**Review:**

**SUMMARY**

This study seeks to improve decoding of object based attention via eye tracking by generating a benchmark dataset and applying decoding techniques to said videos. The MOET dataset is focused on crowded scenes and has multiple repeats with individuals attending different objects which should prove useful in training similar models. The authors then propose a pair of decoding models which utilize gaze density maps across several frames, but ultimately underperform relative to simpler benchmark models. This study provides a roadmap for improved decoding of overt attention.

**Originality**
- The generated MOET dataset provides a novel set of eye-tracking data. Importantly, participants tracked different objects across several repeats across the same video (which turns out to be very useful in decoding overt attention).
- The proposed models build off of prior proposals with their gaze-pooling layer but is novel to the particular task of attention decoding.

**Quality**
- Overall, both the experiment and modeling were well explained. The eye tracking experiment was generally well designed and takes on the challenging goal of having individuals track moving objects in the presence of many other moving objects.

*Weaknesses*
- The task design feels a bit awkward given the eventual objective of being able to decode overt attention from eye-gaze. Participants are instructed to attend to an object indicated by an ellipse (note figure 1 shows a bounding box, clarify, we should be able to infer what participants see) and frames are subsequently excluded if their gaze moves too far from the bounding box. It would seem like a participant would likely be attending the box itself rather than the object contained within. This would result in very different eye movements than if an individual was spontaneously tracking an object and may limit the performance of any resulting decoder on other datasets. This is a challenging/impossible thing to disentangle. The authors should discuss, explain the rationale and any subsequent limitations of decoders trained of this dataset.
- Separately, the exclusion criteria feel a bit arbitrary and potentially counter to the fields objectives (decoding overt attention from natural videos). What does 100 pixels correspond to in terms of degrees of visual angle? Excluding frames where the participant is likely not completing the task is fine, but a rationale should be provided. Should report which % of objects contained frames dropped for this reason (instead of just the total number). Depending on how these are distributed, an alternative approach would be to instead drop an entire tracked object if any (or n) frames fall beyond a certain threshold but otherwise maintain the whole sequence.
- One obvious weakness of the proposed decoder models is that they do not have access to the initial bounding boxes used in generating the task. It would be nice to see a model that builds on the baseline models but uses the gaze density map to get a smoother estimate.

**Clarity**
- The paper was generally quite clear/ easy to follow.

*Weaknesses*
- Figure 1, it would be useful to show examples of frames that would be dropped because gaze is too far from object.
- Details surrounding data collection are missing. How big and far away is the screen? How big is the screen in degrees of visual angle?

**Significance**
- The dataset provided could prove to be a very valuable benchmark to compare future work in this space.
- The performance of the proposed models is a bit disappointing (especially considering the feedforward model beats out the history aware recurrent model). That said, the baseline models compared against are quite strong and this provides a great starting point for others to build on.

**Questions**
- The analysis presented in tables 3 and 4 feel very arbitrary and the reason for specifically training on videos 1-4 is not obvious to the reader. What is the rationale for this subsetting? Or even better can we just see the full train-test performance matrix?
- Related, the performance of the Feedforward and baseline models feels relatively disappointing. As such the cross decoding results in tables 3-4 may not be super informative on its own. Are the videos that are easier to decode with the feedforward model also easier for the baseline oracle or object detector models? It would be more helpful to present performance relative to these models than raw performance, or rather just present performance of one of these baseline models across videos too.
- Objective measures of data quality. How good are individuals at tracking objects in general? What is the average % of time where participants are within the objects bounding box? What is the distribution of deviations when they are outside of it? Sharing these metrics would prove useful in assessing participant performance and comparing with other datasets.

**Overall**

Good quality benchmark dataset on gaze tracking. More details needed on collection and better/ more complete decoding metrics needed.

---

### Official Review · Reviewer_nEQ3 · 2022-10-19
**Benchmark dataset and models for attention encoding**

**Rating:** 8
**Confidence:** 4

**Review:**

This paper introduced a new eye tracking dataset (MOET) that is publicly available for gaze tracking tasks. The methodology on the data collection and two end-to-end deep learning models for attention decoding are presented and comparisons with the other state-of-the-art methods are presented.
The main contribution of the new eye gaze tracking data is that it contains video from crowed real-world data, hence more similar to the natural real-world applications.
Using this gaze data as a benchmark data, the authors are able to propose two slightly different deep learning models for attention decoding. Based on the experiments, the authors also provided interesting hypotheses on what makes it difficult to generalize the attention decoding.
Overall, this paper is well written. The contribution is clear and the methodology on data collection seems to be comprehensive. I would recommend acceptance of this paper.
Some comments:
1.	The experiments compared the generalization ability across participants vs across video. Is it possible to see whether the generalization ability also differs by the targeted objects? In another word, does gaze attention map change with respect to the current tracking subjects?
2.	Line 306: there might be some typo. The sentence doesn’t read correct.
3.	Object Detector (OD) Mod method: The definition of “nearest bounding box” seems a bit ad-hoc. When using only the nearing pixel has the underlying risk of ignoring the scale /size of the bounding box.

---

### Meta-Review · Area_Chair_upQb · 2022-10-20

**Recommendation:** Accept (Poster)
**Confidence:** 5

**Metareview:**

This paper proposes a dataset of eye-tracking for overt attention built on the existing Multiple Object tracking 2016 dataset. The work details the data collection, the protocols, and the post-processing to prepare the data.

The reviewers agree that the dataset is useful and make several useful suggestions to clarify some of the design choices in the data collection. They recognize that while the proposed models did not outperform the strong baselines, the overall dataset is useful.

I recommend an acceptance.

---

### Decision · Program_Chairs · 2022-10-20

Accept (Poster)